# PCB defect detection based on pseudo-inverse transformation and YOLOv5

**Xiaoli Wang**[iD][1,2]*, **Siti Sarah Maidin**[2], **Malathy Batumalay**[2]

**1** Artificial Intelligence Department, Shanxi Polytechnic College, Taiyuan, China, **2** Faculty of Data Science and Information Technology, INTI International University, Nilai, Malaysia

* wangxiaolisxzy@163.com

**Data Availability Statement:** All relevant data are within the paper and its Supporting Information files.

**Funding:** The author(s) received no specific funding for this work.

## Abstract

With the development of integrated circuit packaging technology, the layout of printed circuit boards has become complicated. Moreover, the traditional defect detection methods have been difficult to meet the requirements of high precision. Therefore, in order to solve the problem of low efficiency in defect detection of printed circuit boards, a defect detection method based on pseudo-inverse transform and improved YOLOv5 is proposed. Firstly, a defect image restoration model is constructed to improve image clarity. Secondly, Transformer is introduced to improve YOLOv5, and the batch normalization and network loss function are optimized. These methods improve the speed and accuracy of PCB defect detection. Experimental verification showed that the restoration speed of the image restoration model was 37.60%-42.38% higher than other methods. Compared with other models, the proposed PCB defect detection model had an average increase of 10.90% in recall and 12.87% in average detection accuracy. The average detection accuracy of six types of defects in the self-made PCB data set was over 98.52%, and the average detection accuracy was as high as 99.1%. The results demonstrate that the proposed method can enhance the quality of image processing and optimize YOLOv5 to improve the accuracy of detecting defects in printed circuit boards. This method is demonstrably more effective than existing technology, offering significant value and potential for application in industrial contexts. Its promotion could facilitate the advancement of industrial automation manufacturing.

## 1. Introduction

With the continuous development of industrial automation manufacturing, integrated circuit packaging technology (ICPT) has gradually become the core of global economic development. Among them, printed circuit board (PCB), as an important electronic component in ICPT, undertakes the support and line connection of electronic components [1,2]. However, the rapid development of ICPT has resulted in more complex and dense PCB circuit design. Traditional PCB defect detection (PCB-DD) methods have been difficult to meet the fine PCB layout and high precision requirements of manufacturing companies. Automated PCB-DD has new possibilities due to the development of intelligent technologies such as deep learning (DL) [3].

To solve the low effectiveness of PCB object detection (OD) based on deep neural networks (DNN), C. J. Li et al. developed a PCB-DD approach based on extended feature pyramid

**Competing interests:** The authors have declared that no competing interests exist.

network model. The results revealed that the average accuracy (AP) of the publicly available PCB dataset was 96.2% [4]. To overcome the problem of defects caused by pins in PCBs being fixtured, M. Jeon et al. proposed a non-contact inspection method based on thermal image comparison and DL analysis. Real-time detection and localization of high-precision PCB defective parts was achieved by using structural similarity index maps for regular OD and feature regions of convolutional neural network (CNN) and autoencoder for analysis [5]. A DNN-based PCB detection model was presented by M. A. Alghassab et al. to increase the effectiveness of image processing methods in PCB-DD. In a publicly available PCB dataset, feature extraction and classification were carried out by pre-training CNNs, yielding an accuracy of 94.11% for annoyance [6]. An Atlas spatial pyramid pooling-balanced feature pyramid network was proposed by N. Zeng et al. to enhance tiny target identification performance. Through contextual information linkage, the study enhanced the detection efficiency of tiny surface defects on PCBs by employing Atrus convolution operators with varying expansion rates [7]. M. Yuan et al. proposed an improved you only look once version 5 (YOLOv5) network for the challenge of accurately identifying tiny defects on the surface of PCBs in complex contexts. To achieve an AP of 98.6% in the PCB public dataset, the study enhanced the network's feature extraction capability by using HorNet and improved the attention module and sampling layer of YOLOv5 by using multiple convolutional block attention module and content-aware feature reorganization [8].

YOLOv5, a computer vision target detection DL model, is one of the you only look once (YOLO) series that can achieve accurate and efficient real-time target detection [9]. Scholars in related fields have examined OD research from a variety of angles. Z. Liu et al. proposed an enhanced OD method for the YOLOv5 UAV shooting scene, aiming to address the issues of a large number of dense small objects in high-altitude shooting and the complicated background noise interference in the shooting scene. The network's detection effectiveness for medium- and long-range objects was increased by convolutionally generating adaptive weights for various sensory field features and improving spatial pyramid pooling for feature augmentation [10]. A millimeter-wave radar and vision fusion method was proposed by Y. Song et al. for OD in response to the low efficacy of multi-sensor fusion. To maximize the utilization of sensor information and improve detection accuracy (DA), the radar data was processed using a mapping transform neural network based on YOLOv5, bringing the radar and visual data to the same scale [11]. M. Sadiq et al. proposed a combined region-based CNN and YOLOv5 for safety helmet detection for construction site workers. A robust model was constructed by fusing YOLOv5 and fuzzy image enhancement, which led to separate detection of safety helmets from other types of helmets and was applied in real-world scenarios [12]. The training procedure is hampered by the common branching of the YOLOv5 head's classification and regression tasks, as well as by flaws like the poor association between the classification score and localization accuracy. To address this problem, H. Wan et al. proposed the intersection over union (IoU) perceptually decoupled head for improving YOLOv5. The study significantly improved the model's localization accuracy for real-time OD [13].

Combined with the above, it can be concluded that related scholars have conducted various researches on PCB-DD, and the DL-based defect detection (DD) method has also been widely applied in PCB-DD. Nevertheless, PCB introduces some noise during the picture capture process, which results in lost feature information when combining features based on anchor frames. Moreover, YOLOv5 is prone to ignore the C5 feature map (size: 20×20, channel: 1024) in the feature map generation process of tiny target detection. To improve the PCB-DD performance, the study suggests a pseudo-inverse transform (PIT) and a PCB-DD approach based on YOLOv5 in order to overcome the aforementioned issues and challenges. The study first innovatively proposes a PIT-based image restoration (IR) model for PCB defective images and

improves YOLOv5 using Transformer. At the same time, the normalization processing layer and loss function of YOLOv5 are improved to enhance the DA of the PCB-DD model.

An IR model based on PIT is developed to improve the clarity of PCB defect images. By combining total variation (TV) and shear transformation (ST), this model effectively removes image noise and preserves important details. At the same time, Transformer is introduced to improve YOLOv5, which enhances the ability to extract model features, thus improving the ability to identify PCB defects. By optimizing the batch normalization (BN) and loss function in YOLOv5, the DA of the model is improved. The PCB-DD method proposed in this study has important application value and significance in the actual PCB-DD field, which is conducive to improving the efficiency and accuracy of PCB-DD in the manufacturing industry. The research not only advances the theoretical development of PCB-DDtechnology but also demonstrates clear practical advantages, offering an efficient and precise solution for the detection of defects in industrial automation manufacturing.

## 2. Methods and materials

To realize high-precision PCB-DD, the study firstly carries out PCB image restoration (PCB-IR) algorithm design on the basis of PIT. Secondly, Transformer is used to improve YOLOv5, and the normalization processing layer and loss function of YOLOv5 are improved, so as to construct the improved Transformer-YOLOv5 (T-YOLOv5) PCB-DD model. Finally, the production of PCB-DD dataset is carried out.

### 2.1. PCB image restoration model based on PIT

TV model is commonly used in image preprocessing for PCB-DD, which recovers the image by filtering and regularization [14]. Among them, filtering mainly removes the noise in the image by using a specific filter function. Whereas, regularization is used for noise removal and detail recovery by constructing a model [15]. The regularization expression formula for the TV model is shown in Eq (1).

$$\min_{p} \frac{1}{2} \|\partial - Bp\|_2^2 + \gamma \|Ap\|_1 \tag{1}$$

In Eq (1), $p$ denotes the image to be recovered and $\partial$ denotes the damaged image. $\frac{1}{2}\|\partial - Bp\|_2^2$ denotes the fidelity term and $B$ denotes the observation matrix. $\gamma$ denotes the regularity term parameter and $\|Ap\|_1$ denotes the TV regularity term. $A$ denotes the discrete gradient operator. When the observation matrix $B$ is irreversible, it leads to poor detail recovery of the image. In addition, a single TV regular term cannot take into account the balance between noise removal and detail preservation. Therefore, the study improves the TV and proposes a TV-ST bi-regularized image restoration (TSBRIR) method. Firstly, the study improves the TV regularization using ST to construct a PCB-IR model. The specific objective function (OF) expression is shown in Eq (2).

$$f(p) = \frac{1}{2} \|\partial - Bp\|_2^2 + \gamma_1 \|Ap\|_1 + \gamma_2 \sum_{r=1}^{N} \|C_r p\|_1 \tag{2}$$

In Eq (2), $f(p)$ is the OF of the PCB-IR model. $\gamma_1$ and $\gamma_2$ are the parameters of the TV regular term and ST regular term, respectively. $N$ denotes the total number of ST subbands at $p$ non-downsampling. $r$ denotes the number of ST subbands at $p$ non-downsampling. $C_r$ denotes a block cyclic matrix. $\sum_{r=1}^{N} \|C_r p\|_1$ denotes an ST regular term. Moreover, PIT is introduced to improve the TV fidelity term to obtain the expression formula of the PCB-IR model. In mathematics, if a matrix

with the same number of rows and columns is reversible, it has an inverse matrix. Therefore, the result of multiplying the matrix with its inverse matrix is the identity matrix. However, for non-square matrices or irreversible square matrices (such as singular matrices), the inverse cannot be obtained directly. At present, PIT can be used to solve this problem. PIT is not only applicable to irreversible matrices, but also to invertible matrices. Specifically, it is shown in Eq (3).

$$
\begin{cases}
\min_{p,\tilde{\partial}} \dfrac{1}{2(\vartheta_a + \wp)^2} \|\tilde{\partial} - p\|_2^2 + \gamma_1 \|Ap\|_1 + \gamma_2 \sum_{r=1}^{N} \|C_r p\|_1 \\
\tilde{\partial} = B^\dagger \partial \\
s.t. \quad B\tilde{\partial} = \partial
\end{cases}
\tag{3}
$$

In Eq (3), $\tilde{\partial}$ denotes an auxiliary variable and $B^\dagger$ denotes the pseudo-inverse of the observation matrix $B$. $a$ denotes a vector of Gaussian random variables with mean 0, and $\vartheta_a$ denotes the standard deviation. $\wp$ denotes a parameter that avoids a standard deviation of 0. $\frac{1}{2(\vartheta_a+\wp)^2} \|\tilde{\partial} - p\|_2^2$ denotes the pseudo-inverse fidelity term. For solving the constrained minimum optimization Eq (3), the study solves the OF of the PCB image restoration model based on the framework of the split Bregman (SB) algorithm and uses the adaptive threshold function to compute the sub-problems corresponding to the TV regular terms in the solution process. As an optimization algorithm, the SB algorithm is commonly used to solve sparse problems with global convergence property and desirable convergence speed [16,17]. The auxiliary variables $b$ and $\omega_r$ are first introduced to improve Eq (3) such that $b = Ap$, $\omega_r = C_r p$. Specifically as shown in Eq (4).

$$
\min_{p,\tilde{\partial},b,\omega_r} \dfrac{1}{2(\vartheta_a + \wp)^2} \|\tilde{\partial} - p\|_2^2 + \gamma_1 \|b\|_1 + \gamma_2 \sum_{r=1}^{N} \|\omega_r\|_1
\tag{4}
$$

Next, the Bregman distances $d_k$ and $(d_r)_k$ are introduced to convert Eq (4) to what is shown in Eq (5).

$$
(p_{k+1}, (\omega_r)_{k+1}, b_{k+1}, \tilde{\partial}_{k+1}) = \arg\,\min_{p,\tilde{\partial},b,\omega_r} \dfrac{1}{2(\vartheta_a + \wp)^2} \|\tilde{\partial} - p\|_2^2
$$
$$
+\gamma_1 \|b\|_1 + \gamma_2 \sum_{r=1}^{N} \|\omega_r\|_1 + \dfrac{\wp_1}{2} \|b - Ap - d_k\|_2^2
$$
$$
+\dfrac{\wp_2}{2} \sum_{r=1}^{N} \|\omega_r - C_r p - (d_r)_k\|_2^2
\tag{5}
$$

In Eq (5), $\wp_1$ and $\wp_2$ denote the parameters that prevent the standard deviation from being zero. $k$ is the images. Solving Eq (5) in the framework of the SB algorithm transforms the optimization problem of minimizing the OF of the improved model into four subproblems, as shown in Eq (6).

$$
\begin{cases}
p_{k+1} = \arg\,\min_p \dfrac{1}{2(\vartheta_a + \wp)^2} \|\tilde{\partial} - p\|_2^2 + \\
\dfrac{\wp_1}{2} \|b - Ap - d_k\|_2^2 + \dfrac{\wp_2}{2} \sum_{r=1}^{N} \|\omega_r - C_r p - (d_r)_k\|_2^2 \\
(\omega_r)_{k+1} = \arg\,\min_{\omega_r} \gamma_2 \|\omega_r\|_1 + \dfrac{\wp_2}{2} \|(\omega_r)_k - C_r p_{k+1} - (d_r)_k\|_2^2 \\
b_{k+1} = \arg\,\min_b \gamma_1 \|b\|_1 + \dfrac{\wp_1}{2} \|b - Ap_{k+1} - d_k\|_2^2 \\
\tilde{\partial}_{k+1} = \arg\,\min_{\tilde{\partial}} \dfrac{1}{2(\vartheta_a + \wp)^2} \|\tilde{\partial} - p_{k+1}\|_2^2
\end{cases}
\tag{6}
$$

In Eq (6), $p_{k+1}$ denotes the transformed subproblem 1. $(\omega_r)_{k+1}$ denotes the transformed subproblem 2. $b_{k+1}$ denotes the transformed subproblem 3. $\tilde{\partial}_{k+1}$ denotes the transformed subproblem 4. Based on the four subproblems obtained from the transformation, a step-by-step solution is performed. First, subproblem 1 $p_{k+1}$ is solved by taking the derivative of the right-hand side of the equation and making it equal to 0, as shown in Eq (7).

$$(I + (\wp_2 * (\vartheta_a + \wp)^2)A^T A + (\wp_2 * (\vartheta_a + \wp)^2)\sum_{r=1}^{N} C_r^T C_r)p = \tilde{\partial}_k + (\wp_2 * (\vartheta_a + \wp)^2)A^T(b_k - d_k)$$

$$+ (\wp_2 * (\vartheta_a + \wp)^2)\sum_{r=1}^{N} C_r^T((\omega_r)_k - (d_r)_k) \tag{7}$$

In Eq (7), the matrix transpose is denoted by $T$ and the unit matrix is denoted by $I$. The study utilizes the conjugate gradient method to solve Eq (7), and the solution is the value of subproblem 1 $p_{k+1}$. Next, subproblem 2 $(\omega_r)_{k+1}$ is solved using the soft threshold function. The specific expression is shown in Eq (8).

$$(\omega_r)_{k+1} = shrink(C_r p_{k+1} + (d_r)_k, \frac{\gamma_2}{\wp_2}) \tag{8}$$

In Eq (8), $shrink(\cdot)$ denotes the soft threshold function. On this basis, solve subproblem 3 $b_{k+1}$. Since subproblem 3 is a subproblem about denoising the TV regular term, and the soft threshold function processed values will produce errors with the original values, it is investigated to introduce a nonlinear function for asymptotic compression of the threshold value to remove the errors [18]. The specific formula is shown in Eq (9).

$$\begin{cases} (b_{k+1})_i = \Psi((Ap_{k+1} + d_k)_i, \frac{\gamma_1}{\wp_1}) \\ \\ \Psi(\varpi, w) = \begin{cases} sign(\varpi) \times (|\varpi| - w + \dfrac{w}{2\lambda + 1}), |\varpi| \geq w \\ \\ \dfrac{(\varpi)^{2\lambda+1}}{(2\lambda + 1) \times (w)^{2\lambda+1}}, |\varpi| < w \end{cases} \end{cases} \tag{9}$$

In Eq (9), $\lambda$ denotes the regulating factor and $(*)_i$ denotes a definite element in $*$. $\Psi(\cdot)$ denotes the nonlinear function, and $\varpi$ and $w$ denote the two input vectors in the nonlinear function. $sign(\cdot)$ denotes the symbolic function. To achieve better denoising effect, the dynamic adjustment of the threshold function value can be realized by adjusting $\lambda$. At the same time, the subproblem 4 $\tilde{\partial}_{k+1}$ is solved. The specific expression formula is shown in Eq (10).

$$\tilde{\partial}_{k+1} = B^\dagger \partial + (I - B^\dagger B)p_{k+1} \tag{10}$$

After solving all the subproblems, the Bregman distance is updated and the cutoff condition is set. The specific expression formula is shown in Eq (11).

$$\begin{cases} d_{k+1} = d_k + Ap_{k+1} - b_{k+1} \\ (d_r)_{k+1} = (d_r)_k + C_r p_{k+1} - (\omega_r)_{k+1} \\ \|p_{k+1} - p_k\|_2^2 / \|p_k\|_2^2 < \theta \end{cases} \tag{11}$$

In Eq (8), $\theta$ denotes the cutoff condition. Combining the above, when the TV regular term is anisotropic TV or isotropic TV, the computational flow is shown in Fig 1.

In Fig 1, the SB algorithm is first initialized and the parameters, conditioning factors, and cutoff conditions for model solving are set. Secondly, it solves the four subproblems of the transformation and updates the Bregman distance. When the solved subproblems satisfy the

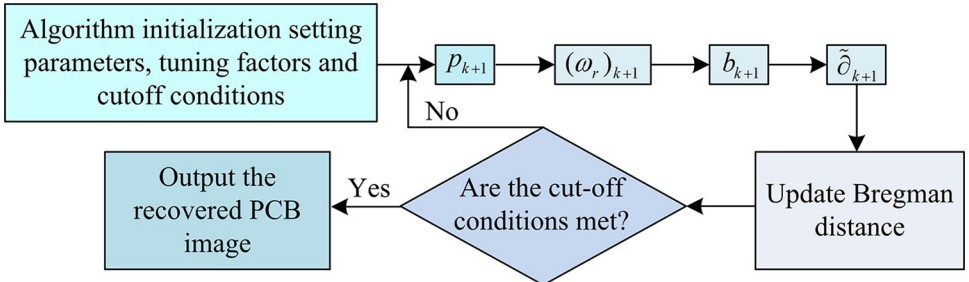

**Fig 1. Pseudo-inverse transform-based PCB image restoration model solving process.**

cutoff condition, the output of PCB recovery image is performed. On the contrary, the 4 sub-problems solving is repeated in order to find the optimal solution. It is worth mentioning that when the TV regular term is anisotropic TV or isotropic TV, the expression of subproblem 1 is updated as shown in Eq (12).

$$(I + (\wp_2 * (\vartheta_a + \wp)^2)[(A^o)^T A^o + (A^q)^T A^q] + (\wp_2 * (\vartheta_a + \wp)^2)\sum_{r=1}^{N} C_r^T C_r)p =$$

$$\tilde{\partial}_k + (\wp_2 * (\vartheta_a + \wp)^2)(A^q)^T((b^q)_k - (d^q)_k) + (\wp_2 * (\vartheta_a + \wp)^2)(A^o)^T((b^o)_k - (d^o)_k)$$

$$+(\wp_2 * (\vartheta_a + \wp)^2)\sum_{r=1}^{N} C_r^T((\omega_r)_k - (d_r)_k) \tag{12}$$

In Eq (12), $A^o$ denotes the horizontal gradient and $A^q$ denotes the vertical gradient. $o$ and $q$ denote the conjugate gradient, respectively. In addition, when the TV regular term is isotropic TV, subproblem 3 also needs to be further updated as shown in Eq (13).

$$\begin{cases} ((b^q)_{k+1})_i = \Psi\left((\kappa)_i, \dfrac{\gamma_1}{\wp_1}\right)\dfrac{A^q p_{k+1} + (d^q)_k}{\kappa} \\ ((b^o)_{k+1})_i = \Psi\left((\kappa)_i, \dfrac{\gamma_1}{\wp_1}\right)\dfrac{A^o p_{k+1} + (d^o)_k}{\kappa} \end{cases} \tag{13}$$

In Eq (13), $\kappa$ denotes the square root of the sum of squares of the conjugate gradients. Although Fourier transform and wavelet based methods are very effective in image processing, they usually need to make a global assumption about the image degradation process, such as assuming that the image degradation is linear and shift invariant. PIT can be directly adapted to specific degradation models, such as fuzzy kernel, making it more effective in dealing with specific types of degradation. The degradation characteristics of PCB images vary due to differences in manufacturing processes and conditions. PIT is capable of adapting to these changes because it is not based on a fixed transformation basis but rather on the specific information of images and the degradation process itself. Consequently, the application of PIT to enhance the fidelity term of the TV model ensures the preservation of image details and texture information, while simultaneously facilitating the removal of noise during the restoration process. This method demonstrates significant advantages in enhancing IR speed and quality.

## 2.2. PCB-DD model based on PIT and YOLOV5

According to the PCB-IR model based on PIT proposed in the previous section, the study further carries out the improvement of YOLOv5, which leads to the establishment of PCB-DD model based on PIT and YOLOv5. It is found that the YOLOv5 model in the process of DD of PCB images, the input image (II) will be resized to 640×640 size and downsampled by Fcous

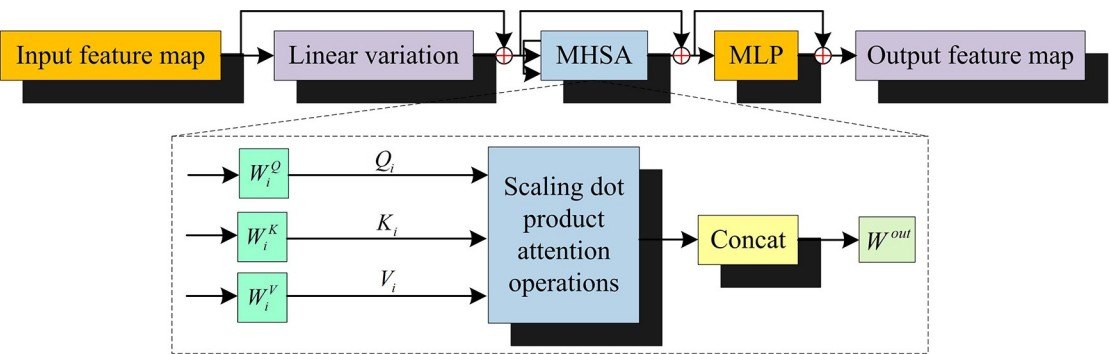

**Fig 2. Schematic diagram of transformer network layer structure.**

module and Conv layer into C1 (size: 320×320, channel: 64), C2 (size: 160×160, channels: 128), C3 (size: 80×80, channel: 256), C4 (size: 40×40, channel: 512) and C5 (size: 20×20, channel: 1024) five feature maps [19–21]. However, multiple downsampling operations result in PCB defect features in the C5 feature map being ignored, which reduces the PCB-DD accuracy. Therefore, the study utilizes the encoder module of Transformer to replace the Bottleneck module of YOLOv5 to improve the YOLOv5 model. Transformer's self-attention mechanism can operate on the whole image and capture the global context information. This is particularly beneficial for elucidating the interrelationships and interactions between disparate objects within a PCB image, thereby facilitating more accurate identification of the relationships between different defects. In addition, multiple downsampling in YOLOv5 can lead to the loss of important information in the high-level feature map. Transformer can supplement this information, and through its encoder-decoder architecture, it can reintegrate the low-level detailed information with the high-level semantic information. The Transformer network layer structure used in the study is shown in Fig 2.

In Fig 2, the dimension of the input feature map matrix $X$ is $[g,e,H,J]$. $g$, $e$, $H$, and $J$ denote the input feature images, the channels, and the size, respectively. $W_i^Q$, $W_i^K$ and $W_i^V$ denote the trainable linear variation matrix. $Q_i$, $K_i$ and $V_i$ denote the corresponding matrices of the linear variation matrix, respectively. $W^{out}$ denotes the linear transformation matrix. The feature image is reconstructed from the matrix consisting of dimensions height and width to obtain the $[H×J,g,e]$ form to comply with the input feature image requirements of multi-head self-attention (MHSA) [22,23]. Second, the feature matrix is combined with the pixel-by-pixel positional coding information of the feature image to obtain a new input vector. The new input vectors are subjected to multi-head self-attentive perception in MHSA, and the reconstructed 2D image is output by the multilayer perceptron, resulting in a feature map containing more global and contextual information. The self-attention potential of MHSA enables the incorporation of more global and contextual information into the flat output feature maps. This results in the input vectors in the MHSA, after multiplication by the trainable linear variation matrix, having the corresponding matrix formulae as shown in Eq (14).

$$\begin{cases} Q_i = UW_i^Q \\ K_i = UW_i^K \\ V_i = UW_i^V \end{cases} \tag{14}$$

In Eq (14), $U$ denotes the input vector. According to the obtained matrix formula, the study utilizes the scaled dot product attention operation to process the matrix and introduces the

Softmax function to constrain the calculation results, so as to obtain the attention score of the corresponding attention head. The specific expression formula is shown in Eq (15).

$$
\begin{cases}
head_i = Attention(Q_i, K_i, V_i) = \text{softmax}(\dfrac{Q_i \cdot K_i^T}{\sqrt{\beta_k}})V_i \\
MultiHead(Q, K, V) = Concat(head_1, \dots, head_i)W^{out}
\end{cases}
\tag{15}
$$

In Eq (15), $head_i$ denotes the attention head of YOLOv5. $MultiHead(\cdot)$ denotes the result of MHSA processing. $Attention(\cdot)$ denotes the attention score. $\beta_k$ denotes the resultant limiting parameter of the Softmax function, and softmax($\cdot$) denotes the Softmax function. $Concat(\cdot)$ denotes extended fusion. Therefore, the C3 structure of the Transformer encoder after improving the Bottleneck module of YOLOv5 is shown in Fig 3.

In Fig 3, in improved C3 (IC3), the input feature image first passes through a 1×1 convolutional layer for channel number compression. The output feature image of the previous layer directly passes through a 3×3 convolutional layer for channel number adjustment and is converted into serialized data to be summed with the position code. After summation, the output feature image is transferred as input to the MHSA for residual concatenation and spliced with the input feature image again to obtain the final output of IC3. In YOLOv5, it has 8 C3 modules and 4 are in Backbone module and 4 are in Preiction module [24]. Therefore, the schematic diagram of YOLOv5 backbone network after improving the C3 modules is shown in Fig 4.

In Fig 4, the study does not replace the shallow C3 module in the T-YOLOv5 backbone network, and only replaces the deeper C3 module with IC3. Since the BN normalization process relies heavily on the mean and method within each batch during the conventional YOLOv5 network training process and ignores the differences in each sample [25,26]. Therefore, the study improves the BN of the network by adding centering and scaling calibration to the beginning and ending parts of the Normalization layer of the network. This is shown in Eq (16).

$$
\begin{cases}
X_{cm} = X + w_m \odot \dfrac{1}{HJ}\sum_{h=1}^{H}\sum_{j=1}^{J}X_{(s,e,h,j)} \\
X_m = X_{cm} - \bar{X}_{cm}
\end{cases}
\tag{16}
$$

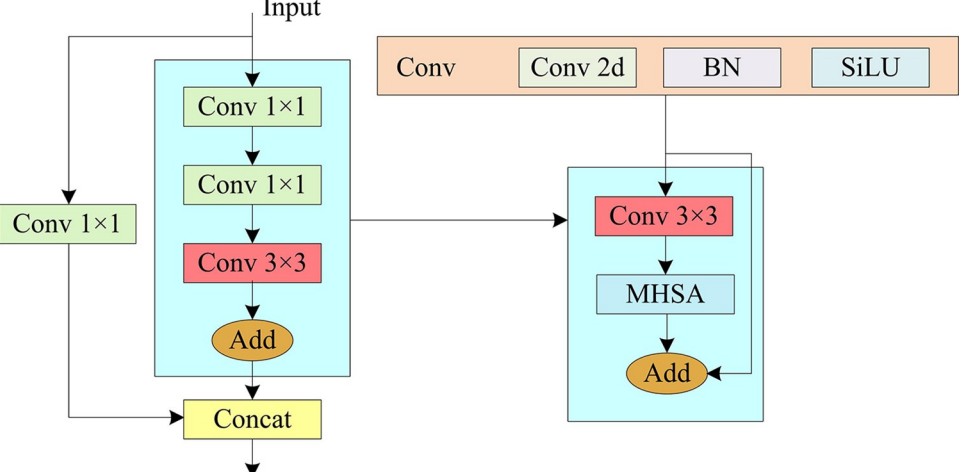

**Fig 3. Schematic of the C3 structure of T-YOLOv5.**

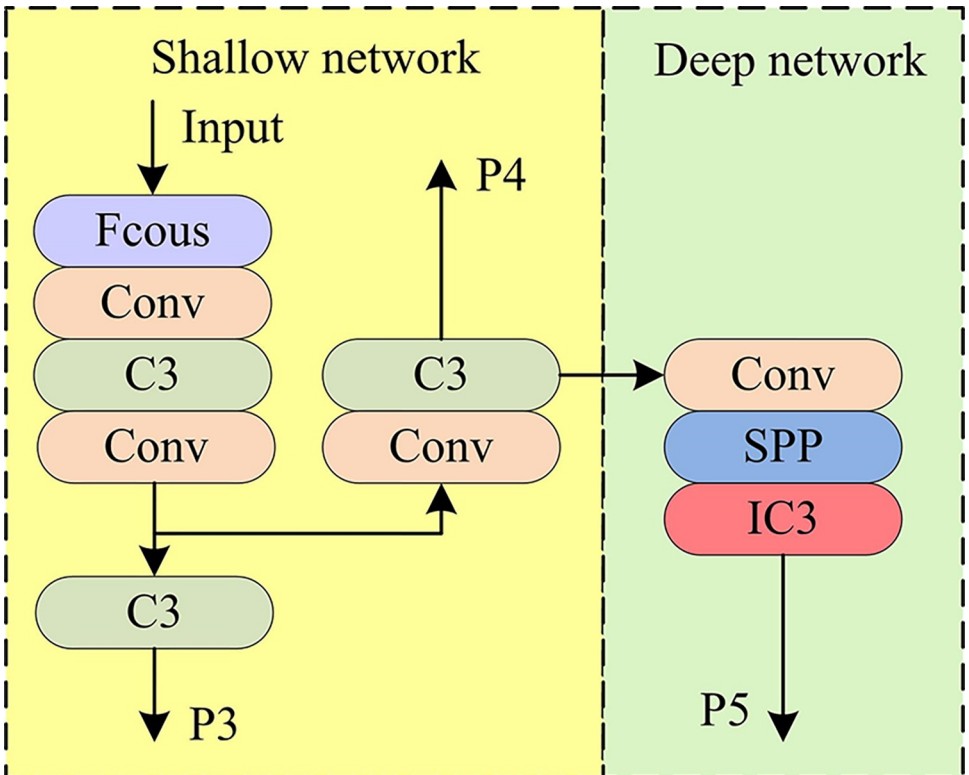

**Fig 4. Schematic diagram of T-YOLOv5 backbone network.**

In Eq (16), $X_{cm}$ denotes the $X$ centering calibration result. $w_m$ denotes the weight vector that can be used for training. $\odot$ denotes the dot product operator. $s \in S$ denotes the input batch size (BS). $\bar{X}_{cm}$ denotes the mean value of $X_{cm}$. $X_m$ denotes the centering feature. The centered features are subjected to a scaling process after the centering calibration is performed and a constant is introduced to avoid the denominator to be 0. Based on this, the Tanh function is utilized to limit the range of extreme values to be taken [27]. Therefore, after the scaling process and training, the final BN results are expressed as shown in Eq (17).

$$\begin{cases} X_s = \dfrac{X_m}{\sqrt{Var(X_m) + c}} \\ X_{cs} = X_s R(w_v \odot K_s + w_g) \\ Y = X_{cs}\varphi + \mu \end{cases} \tag{17}$$

In Eq (17), $X_s$ denotes the centered feature after performing the scaling process. $Var(X_m)$ denotes the $X_{cm}$ variance value. $c$ denotes the constant that avoids $Var(X_m)$ to be 0. $X_{cs}$ denotes the original features obtained after scaling calibration. $R$ denotes the image dimension. $w_v$ and $w_g$ both denote weight vectors that can be used for training. $K_s$ denotes the statistics of $X_s$ and $Y$ denotes the final BN result. $\varphi$ and $\mu$ denote the scaling parameter and the translation parameter, respectively. In addition, the loss function as an important influence on the model performance, the conventional YOLOv5 loss function is mainly composed of three losses: bounding box regression, classification and confidence [28,29]. The specific expression formula is shown

**Table 1. PCB defect types.**

| Small size defects | Large size defects | Hole type defects | Other types of defects |
|---|---|---|---|
| Missing hole | Broken solder joints | Round hole deformation | Circuit breaker |
| Edge lifting | Broken pads | Round hole misalignment | Short circuit |
| Small spot | Copper surplus | Cracks inside the hole | Rodent bite |
| Micro etching | Broken wire | Burr inside round hole | Burr |
| Conductor spacing issue | Layers | Uneven size of holes | Craters |

in Eq (18).

$$Loss = l_{obj} + l_{cis} + l_{box} \tag{18}$$

In Eq (18), *Loss* denotes the loss function and $l_{obj}$ denotes the confidence loss. $l_{cis}$ denotes classification loss and $l_{box}$ denotes bounding box regression loss. In $l_{box}$, generalized IoU (GIoU) is able to distinguish between different real frames and prediction frames. However, when the prediction frame is in the true frame, it degrades to an IoU, which prevents accurate prediction [30]. Therefore, the study utilizes α-IoU loss to replace GIoU as the model training loss function of T-YOLOv5. The specific expression formula is shown in Eq (19).

$$l_{box} = l_{\alpha-IoU} = 1 - IoU^{\alpha_1} + \rho^{\alpha_2}(v, v^{gt}) \tag{19}$$

In Eq (19), $l_{\alpha-IoU}$ denotes the α-IoU Loss function. $\alpha$ denotes the adjustable parameter. $v$ denotes the predicted frame area. $v^{gt}$ denotes the real frame area and $\rho$ denotes the predicted frame. Combining the above, the flow of the PCB-DD model based on PIT and YOLOv5 proposed in the study is shown in Fig 5.

In Fig 5, the study firstly utilizes PIT to improve the TV model and constructs the IR model for PCB-DD in order to improve the clarity of PCB images. ST and biregularization strategies are introduced to perform the image recovery process in the framework of SB algorithm. The OF is transformed into a subproblem that can be solved in steps by introducing auxiliary variables and Bregman distance to output the recovered PCB image. Secondly, Transformer is

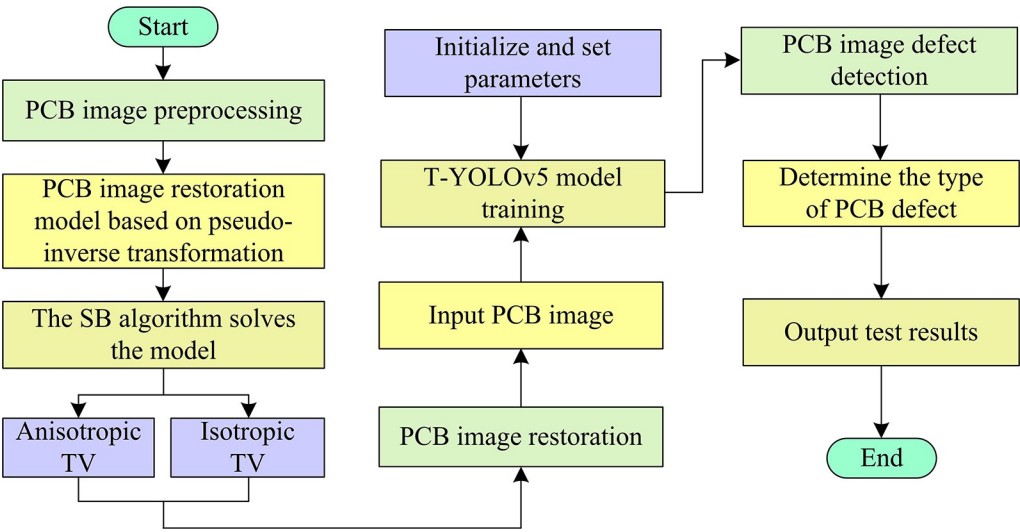

**Fig 5. PCB-DD model flow based on pseudo-inverse transformation and YOLOv5.**

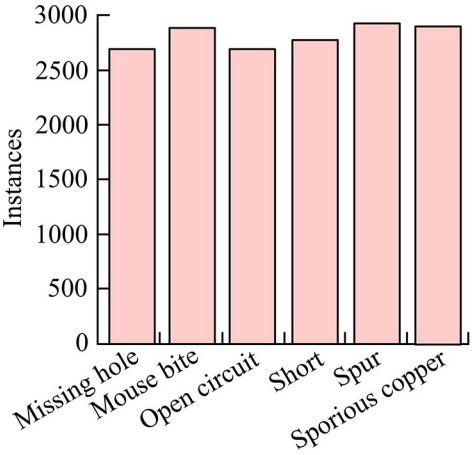

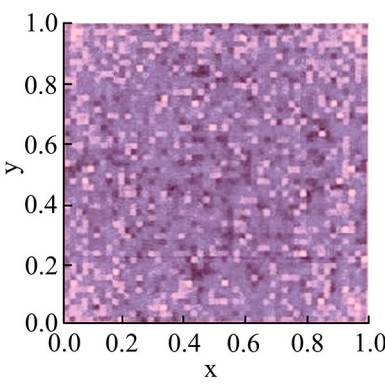

(a) Data set object class distribution          (b) Distribution of center point in object

**Fig 6. Analysis chart of PCB number series.**

introduced to improve YOLOv5, and the BN and loss function of YOLOv5 are optimized. The recovered image is input into T-YOLOv5 for DD and the results are output.

## 2.3. Self-built PCB-DD dataset

Since PCB-DD mainly analyzes the image data of PCBs in order to determine their defects, the criteria for defining PCB defects are not clearly defined. Therefore, the study uses the Institute of Printed Circuit (IPC) standard as the principle of PCB-DD, and uses the PCB open dataset released by Peking University as the basis for data enhancement and other techniques to expand the number of PCB dataset samples. The PCB-DD dataset contains 1386 images, covering 6 different types of defects, including pin holes, mouse bites, open circuits, short circuits, dispersion, and miscellaneous copper. These images can be used for image detection, classification, and registration tasks. The images in the dataset are captured by industrial cameras with a resolution of 1920×1080, and the dataset is divided into training, validation, and testing sets in a ratio of 6:2:2. The format of the dataset follows the PASCAL VOC format. The IPC standards are shown in Table 1.

According to the PCB defect types shown in Table 1, the study flipped, randomly cropped, and added Gaussian white noise to 693 PCB images to obtain 12890 PCB images. The images contain 23464 defects of different types. It mainly includes 3919 hole defects, 3987 rat tear defects, 3855 open circuit defects, 3801 short circuit defects, 3913 burr defects, and 3989 residual copper type defects. For the purpose of training models and validating their performance, the dataset is split 9:1 into training and test sets. The visual analysis of PCB data set is shown in Fig 6.

Fig 6a and 6b show the sample distribution of the PCB dataset and the position of the center of the object. There are 11,601 images in the training set and 1,289 images in the test set. There are 11,601 images in the training set and 1,289 images in the test set.

## 3. Results

Firstly, the effectiveness of the proposed PIT-based PCB-IR model is verified with natural scenery images and PCB images. Secondly, the performance of T-YOLOv5 and its based PCB-DD model is validated using accuracy, recall, mean average precision (mAP) and frame per second (FPS).

### 3.1. PIT-based PCB image restoration model validation

Both PCB and natural light images are used in the study in order to confirm the validity of the PCB-IR model based on PIT. 1.28 and 0.001 are the regularization parameters that are set, respectively. The adjustment factor is set to 1 and the cutoff condition is 0.0001. Meanwhile, two-step iterative shrinkage/thresholding (TwIST) algorithmic framework is introduced to compare with SB algorithmic framework. Moreover, the anisotropic TV calculation under TwIST solution is defined as TwIST-1 and isotropic TV is defined as TwIST-2. The anisotropic TV under SB algorithmic framework solution is defined as SB-1 and isotropic TV is defined as SB-2. To assess the performance of the PCB-IR model under the two solution frameworks, the various methods are applied to four natural light photos with varying values of blur kernel, peak signal-to-noise ratio (PSNR), and image processing time. The study firstly compares the PSNR of the four calculation methods under different values of the fuzzy kernel (FK), as shown in Fig 7.

It can be seen from Fig 7A that the research proposed algorithm has the highest PSNR value among the four images when the value of FK is taken as 1. Fig 7B–7D also show the reasonableness and effectiveness of the research proposed using the SB algorithm as a PCB image restoration model solving algorithm. The research proposed method has the highest PSNR value in all natural light image processing. The average increase in PSNR obtained by SB algorithm for restoration of 4 images is 0.68% when FK is 1. The average increase in PSNR obtained by restoration of 4 images when FK is 2 is 2.43% compared to TwIST. When FK was 4, SB

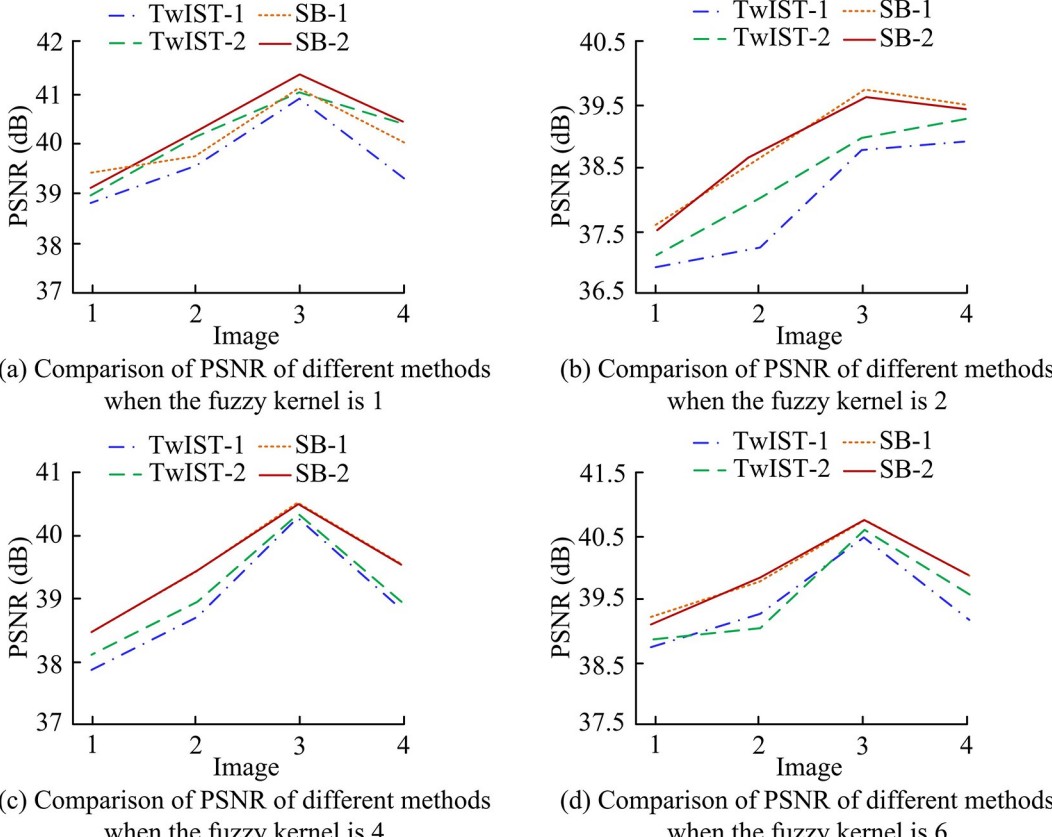

(a) Comparison of PSNR of different methods when the fuzzy kernel is 1

(b) Comparison of PSNR of different methods when the fuzzy kernel is 2

(c) Comparison of PSNR of different methods when the fuzzy kernel is 4

(d) Comparison of PSNR of different methods when the fuzzy kernel is 6

**Fig 7. Comparison of PSNR of different methods with different fuzzy kernels.**

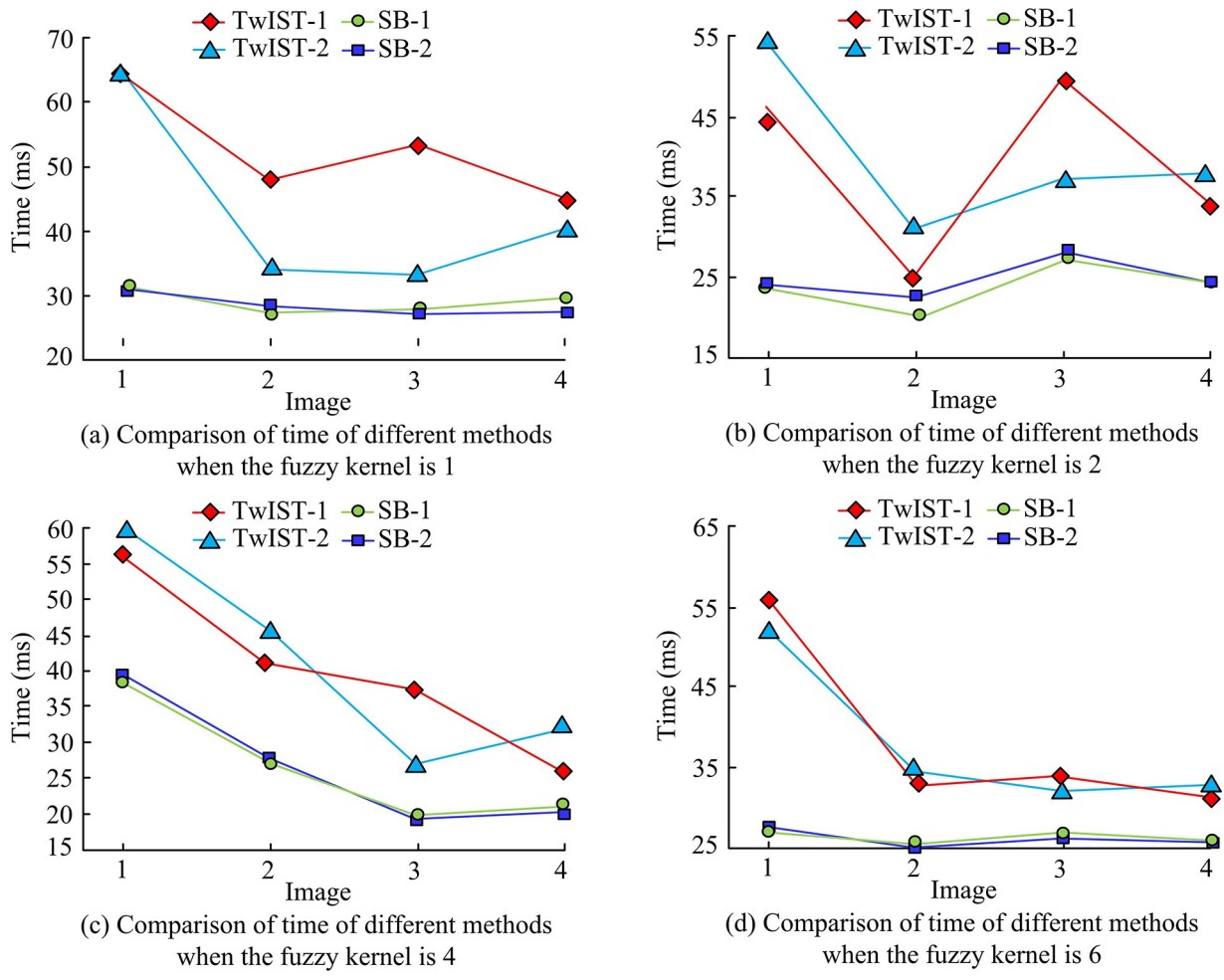

(a) Comparison of time of different methods when the fuzzy kernel is 1

(b) Comparison of time of different methods when the fuzzy kernel is 2

(c) Comparison of time of different methods when the fuzzy kernel is 4

(d) Comparison of time of different methods when the fuzzy kernel is 6

**Fig 8. Comparison of time of different methods with different fuzzy kernels.**

increased PSNR by an average of 1.35% over TwIST. When FK was 6, SB showed an average increase of 1.46%. Among them, the PSNR value of SB-2 is optimal among the four methods, while the PSNR value of TwIST-2 is also significantly better than TwIST-1. This may be due to the fact that isotropic TV can greatly preserve the details of the image and reduce its distortion when restoring the image, thus obtaining a better image quality for restoring the PCB image. TwIST converges slower, often requires more iterations when dealing with large-scale datasets, and it is more sensitive to noise, which may be the reason why TwIST performs inferior to SB. The recovery times of the four methods with different FKs are shown in Fig 8.

From Fig 8A–8D, it can be seen that the time overhead required for the PCB image restoration model proposed in the study is significantly lower than that of TwIST. when FK is 1, the time overhead required for SB is reduced by an average of 39.89% compared to that of TwIST; when FK is 2, the time overhead of SB is reduced by an average of 39.88% compared to that of TwIST; and when FK is 4 and 6, respectively, the time overhead required for SB to in 4 image The time overhead for performing the recovery process is reduced by 38.78% and 40.95% compared to TwIST, respectively. Combining anisotropic TV and isotropic TV also shows that the time overhead of anisotropic TV in performing image restoration is generally lower than that of isotropic TV under the framework of both algorithms, which may be attributed to the fact

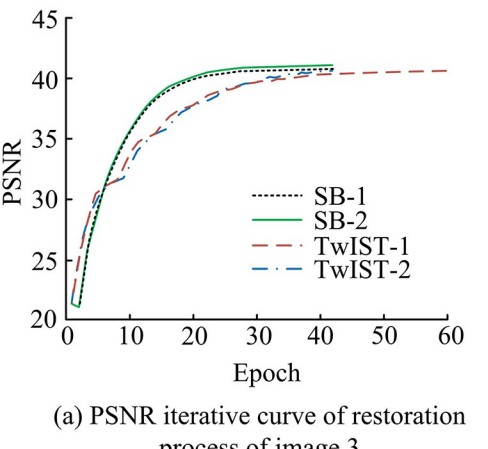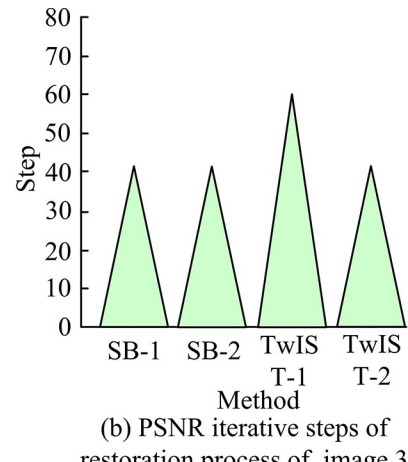

(a) PSNR iterative curve of restoration process of image 3

(b) PSNR iterative steps of restoration process of image 3

**Fig 9. Iterative change of PSNR value during image 3 restoration process.**

that the computational complexity required in solving anisotropic TV is lower, and thus the time it requires is reduced accordingly. However, on the whole, the study proposes that the proposed PCB image restoration model is reasonable and effective using the SB algorithm to solve the proposed PCB image restoration model. At the same time, the study further analyses the iterative change of PSMR values when the four methods perform the restoration process by taking image 3 as an example, as shown in Fig 9.

The study's suggested method's iterative convergence is fastest in Fig 9A during image 3 recovery, whereas the TwIST approach has the slowest convergence and a substantially lower PSNR value than the SB algorithm. Combined with the iteration step size of different methods in Fig 9B, it can be concluded that the SB algorithm is superior. Meanwhile, the study further utilized the acquired PCB degradation images to verify the IR results, which are shown in Fig 10.

Comparison of the PCB degradation image in Fig 10A with the anisotropic TV recovered image in Fig 10B shows that the PCB image is clearer after the recovery. Additionally, the anisotropic TV recovered image in Fig 10C can demonstrate the efficacy of the study's suggested strategy. Therefore, the PCB image recovery model proposed by the study on the basis of PIT can effectively remove the noise and retain the detail information of the PCB image,

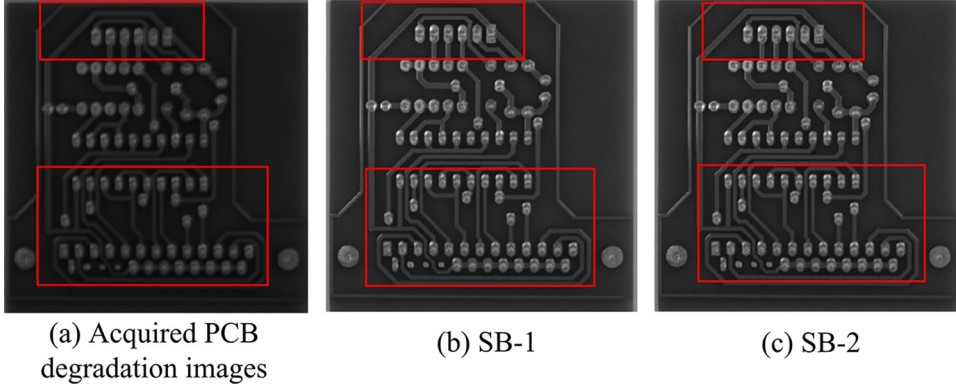

(a) Acquired PCB degradation images

(b) SB-1

(c) SB-2

**Fig 10. Pseudo-inverse transform based PCB image restoration model restoration results.**

**Table 2. Performance comparison of different dataset models.**

| Data set | Accuracy (%) | Precision (%) | Recall (%) | F1 Score (%) |
|----------|--------------|---------------|------------|--------------|
| D1 | 94.57 | 94.53 | 92.33 | 93.42 |
| D2 | 99.20 | 98.97 | 97.87 | 98.42 |

while the model solving in the framework of the SB algorithm is able to obtain a superior PCB recovery image. And in order to further illustrate the applicability of the model in a wider range of scenarios, the study introduces the publicly available PCB-DD (D1) and the self-constructed PCB-DD (D2) from Peking University for the comparison of the model training effect. The details are shown in Table 2.

As can be seen from Table 2, the proposed model has superior training results in the self-built dataset D2, while the model accuracy and F1 score in the unimproved public dataset D1 are above 93%. This indicates that the model is better adapted to the specific features of PCB defect detection and has strong generalisation ability.

## 3.2. PCB-DD model validation based on PIT and YOLOV5

After the proposed PCB dataset is preprocessed by the PIT-based PCB-IR model, the study uses the pre-training weights published by the YOLOv5 developers to train T-YOLOv5. Moreover, the study utilizes stochastic gradient descent (SGD) to update the model parameters. The BS is set to 16, the weight decay is 0.0005, the learning rate is 0.01, the hyperparameter is 0.9, the II size is set to 640×640, and the training rounds is 200. To effectively illustrate the rationality and superiority of the study's improvement of YOLOv5, the study firstly conducts ablation experiments to systematically analyze the system with three parts, namely, IC3, modified BN and α-IoU. Table 3 displays the specific outcomes.

The results of the ablation experiments obtained with IC3, BN and α-IoU when the value of IoU is taken as 0.5 are shown in Table 3. Comparing Experiments 1 and 2, it can be noticed that the improvement of IC3 increased the model mAP value by 1.47%, while the FPS decreased by 3 FPS. The model mAP value of Experiment 3, which only modified the BN, increased by 0.89% compared to Experiment 1. Moreover, the mAP value of the Experiment 4 model with only improved α-IoU increased by 0.50% compared to Experiment 1. This indicates that either improving IC3, BN or α-IoU can improve the performance of the YOLOv5 model. The combination of Experiment 5, Experiment 6 and Experiment 7 further shows that the model mAP value is significantly increased compared to a single improvement. Whereas the combined mAP value of the three improved strategies in Experiment 8 is as high as 99.14%, the mAP values of the other seven experiments increased by 0.32%-2.79%. This

**Table 3. Results of ablation experiments.**

| Experiment number | IC3 | Modify BN | α-IoU | mAP/IoU = 0.5 (%) | FPS |
|-------------------|-----|-----------|-------|-------------------|-----|
| 1 | No | No | No | 96.35 | 46 |
| 2 | Yes | No | No | 97.82 | 43 |
| 3 | No | Yes | No | 97.24 | 49 |
| 4 | No | No | Yes | 96.85 | 45 |
| 5 | Yes | Yes | No | 98.63 | 44 |
| 6 | Yes | No | Yes | 98.82 | 41 |
| 7 | No | Yes | Yes | 98.53 | 46 |
| 8 | Yes | Yes | Yes | 99.14 | 43 |

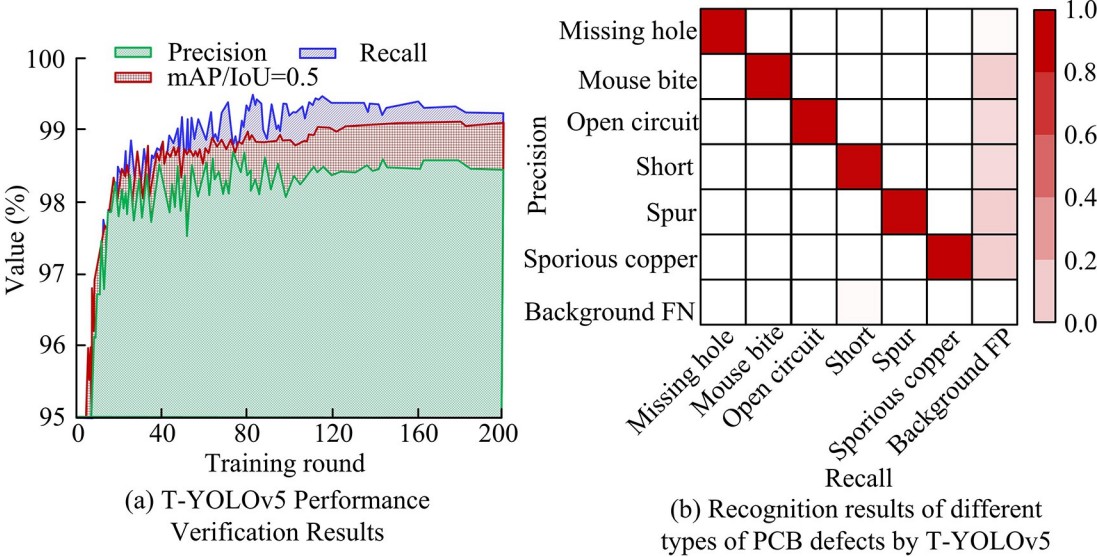

**Fig 11. Validation results of T-YOLOv5 on PCB dataset.**

demonstrates that the study's revised YOLOv5 technique can increase the model's performance. Combining the different methods of improvement shows that IC3 is the main variable that determines the performance of the model. The introduction of MHSA into the IC3 module may be responsible for this improvement. MHSA expands the receptive field of the T-YOLOv5 model, allowing it to capture more global information about the PCB image. Additionally, it addresses the deficiency of T-YOLOv5, which previously resulted in the loss of shallow information when performing continuous downsampling operations. Moreover, the BN modification mainly enhances the feature expression of the T-YOLOv5 model and improves the stability of the image feature distribution. Furthermore, by using α-IoU as the loss function, the degrading faults of GIoU are also eliminated, and the prediction frame is brought closer to the actual frame, improving the accuracy of the loss function's bounding box regression to the target. Based on this, the study goes on to examine the T-YOLOv5 model's performance as well as the findings of its detection of the six PCB dataset fault types. The details are shown in Fig 11.

In Fig 11A, the proposed T0YOLOv5 model of the study has an accuracy of 98.37%, a recall of 99.24%, and a mAP/IoU = 0.5 of 99.15% after 200 rounds of iterations in the PCB dataset. It is evident that it is more successful in identifying the six types of faults when combined with the T-YOLOv5 model in Fig 11B for the six types of PCB-DD confusion matrices. To further illustrate the effectiveness of the PCB-DD model proposed in the study, the study introduces faster region-based convolutional network (Faster R-CNN) [31]. Single shot multibox detector (SSD), YOLOv3, YOLOv4, and the unimproved YOLOv5 are compared with the performance of the proposed T-YOLOv5 [32–34]. Among them, the AP for six defect types of PCB dataset is shown in Fig 12.

In Fig 12, the research proposed T-YOLOv5 has the highest detection results in five types of defects. The AP values in the hole defect types increased by 1.60%-39.19% over the other models. In the rat-bite defect type, the T-YOLOv5 model have an AP value of 98.81%, which is an average increase of 16.28% over the other models. Moreover, the AP value of T-YOLOv5 in the three types of DD, namely open circuit, burr and residual copper, is still 98.52% and above. This indicates the reliability of the study proposing improvements to the YOLOv5 model. The

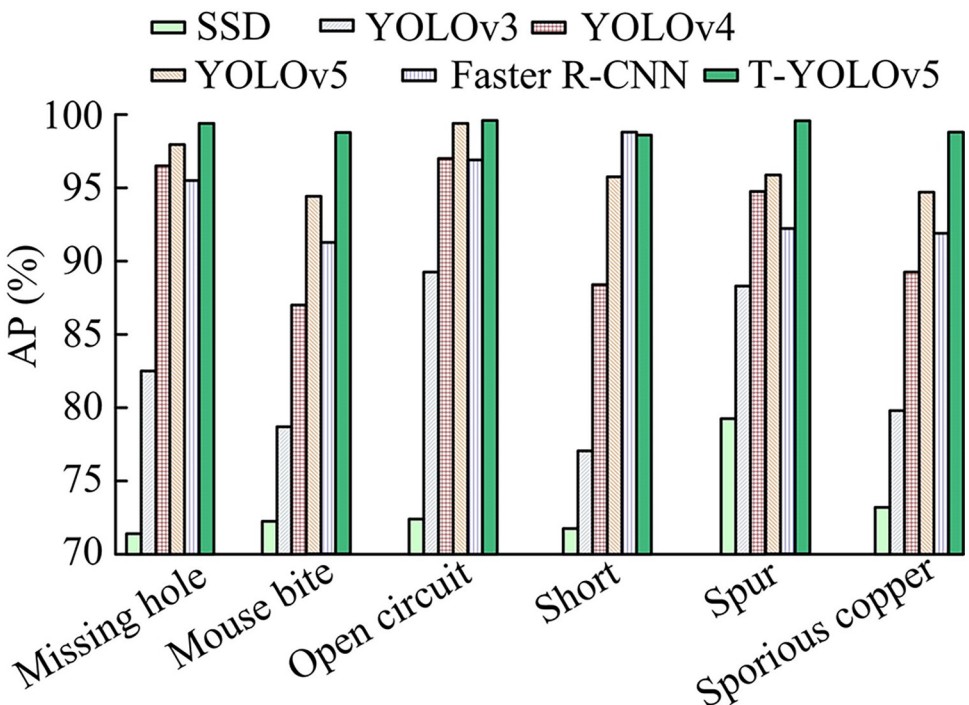

**Fig 12. Absence detection results of different models for each type of absence in the test set of PCB dataset.**

AP value in short-circuit type defects is reduced by 0.11% compared to Faster R-CNN. There may be fewer examples for short-circuit type problems as a result of the training set's unequal sample distribution. Faster R-CNN usually requires higher computational resources and a longer training process, which contributes to the degradation of its DA. And SSD is similar to Faster R-CNN in that its detection results usually need to remove overlapping detection frames through post-processing steps such as non-great value suppression, which increases the complexity of the algorithm. Comparatively speaking, among the six models, the T-YOLOv5 model is more advantageous for PCB-DD. In the meantime, Table 4 presents the precise findings of the study's comparison of the six models' performances in the PCB dataset.

In Table 4, the proposed T-YOLOv5 model of the study has a detection recall of 99.7% in the PCB dataset, which is an average increase of 10.90% compared to the other five models. In terms of mAP value, the T-YOLOv5 model has a mAP of 99.1%, which is an average increase of 12.87% over the other models. The Faster R-CNN model needs to pre-generate candidate ports during the PCB-DD process and then perform sliding window operation with convolutional computation, which may be the reason for its poor FPS and weight. While SSD is mainly recognized by the generated multi-scale feature images, it is weak in recognizing shallow

**Table 4. Comparison of detection results of different models in PCB dataset.**

| Model | Faster R-CNN | SSD | YOLOv3 | YOLOv4 | YOLOv5 | T-YOLOv5 |
|---|---|---|---|---|---|---|
| Backbone network | VGG16 | VGG16 | D3 | C3 | C3 | IC3 |
| Weights (mb) | 298.2 | 105.7 | 225.0 | 246.0 | 28.2 | 28.7 |
| Recall (%) | 96.5 | 79.4 | 84.6 | 91.3 | 97.7 | 99.7 |
| mAP/IoU = 0.5 (%) | 94.4 | 73.4 | 82.7 | 92.3 | 96.3 | 99.1 |
| FPS | 26 | 57 | 39 | 42 | 45 | 43 |

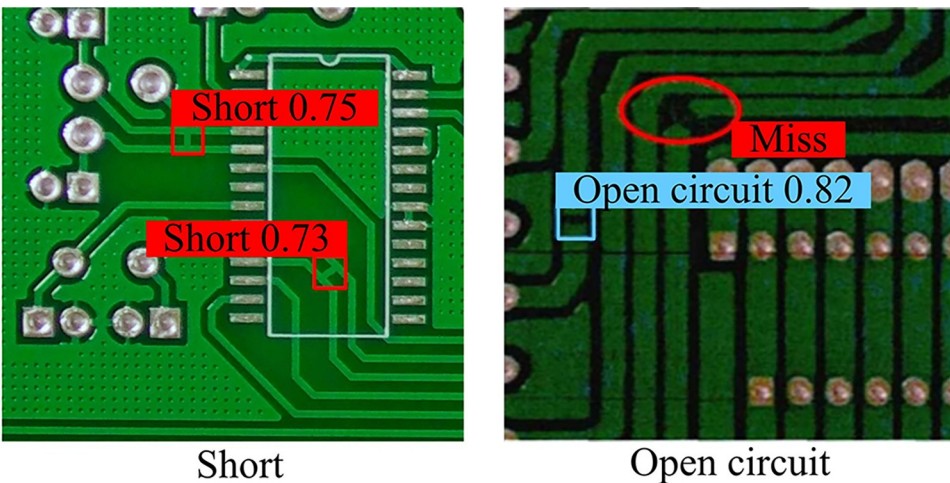

(a) YOLOv5 defect detection results

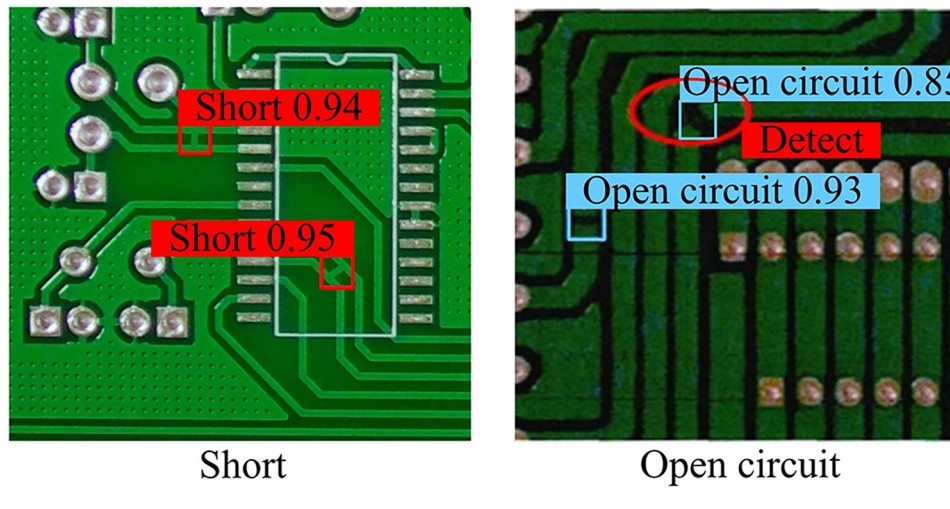

(b) T-YOLOv5 defect detection results

**Fig 13. Comparison of the detection results of the two models for PCB defects "short" and "open circuit".**

feature maps, which may be the reason for its poorer effect on PCB-DD. Further comparing the study's proposed T-YOLOv5 with the YOLO series model, it can be noticed that after IC3, BN, and the loss function, T-YOLOv5 has been significantly improved in terms of detection speed, accuracy, and weight. On this basis, the study further compares the effect of YOLOv5 and T-YOLOv5 in PCB-DD. Fig 13 displays the specific visualization findings.

Comparing the detection results of YOLOv5 in Fig 13A with those of T-YOLOv5 in Fig 13B, it can be observed that the T-YOLOv5 model PCB is more accurate in the detection of open-circuit defects. The combination of the previous validation results provides further evidence of the effectiveness of the PCB-DD model based on PIT and YOLOv5, as proposed by the study. Additionally, the study's improvement of YOLOv5 has been demonstrated to enhance the DA of the PCB-DD model. Finally, the DNN-based PCB-DD method proposed in reference [4] and the CNN-based PCB-DD method proposed in reference [6] are presented and compared with T-YOLOv5. Based on these literatures, the study proposes optimal

**Table 5. Performance comparison of different methods.**

| Model | Recall | | Testing time | | Accuracy | |
|---|---|---|---|---|---|---|
| | *t*-value | *p*-value | *t*-value | *p*-value | *t*-value | *p*-value |
| T-YOLOv5 vs DNN | 2.35 | <0.001 | 3.45 | <0.001 | 2.41 | <0.001 |
| T-YOLOv5 vs CNN | 4.56 | <0.001 | 4.87 | <0.001 | 3.59 | <0.001 |

parameters. The performance is compared and the test is repeated five times to get the t-test results as shown in Table 5.

In Table 5, the *t*-values between the proposed method and other methods are all above 1.96 (at 95% confidence level, two-tailed test). It shows that the differences between different methods are statistically significant. Concurrently, the results illustrate the efficacy and superiority of the proposed method.

## 4. Conclusion

To improve the effectiveness of the current detection model in image DD of PCB, the study proposed a PCB-DD model based on PIT and YOLOv5. Firstly, the PIT was utilized to improve the TV model to construct the PCB-IR model, and the SB algorithm was utilized to solve the model. On this basis, TOLOv5 was further improved using Transformer, and the network BN and loss function were optimized, so as to construct a PCB-DD model on the basis of T-YOLOv5. The experimental validation shows that the PSNR values of SB recovered images are increased by 0.68%, 2.34%, 1.35%, and 1.46% over TwIST when FK is taken as 1, 2, 4, and 6, respectively. The accuracy of the proposed T-YOLOv5 model in the study after 200 rounds of iterations in the PCB dataset was 98.37%, the recall was 99.24%, and the mAP/IoU = 0.5 was 99.15%. Compared to the other models, the T-YOLOv5 model achieved an average increase in recall of 10.90% and an average increase in mAP value of 12.87%. The results demonstrated that the construction of a PCB-IR model using PIT could enhance the quality of the image. Furthermore, the improvement of YOLOv5 on this basis could enhance the DA of the model. The PCB-DD model proposed in the study was capable of retaining image detail information and improving the acquisition of shallow image features, thereby enhancing the recognition and detection of PCB image defects. This had the potential to have a positive impact in the field of PCB-DD. By more accurately detecting defects in PCBs, it is possible to reduce the number of defective products reaching the market, thereby improving the reliability and quality of the final product. Automated and efficient defect detection can reduce the cost of manual inspection and potentially reduce material waste and rework due to defects. As electronic devices become more complex, PCBs need to be designed and manufactured with greater precision. This technology can support more complex and detailed PCB designs. In addition, fast and accurate defect inspection can shorten the time it takes to get a product from design to market. Nevertheless, a limitation of the study is that the design of a visual system for PCB defect recognition has not been further developed. In the future, the proposed model will be considered to be optimized by combining client-side, optimization algorithms, etc., and a PCB visual DD system will be developed. In addition, the T-YOLOv5 model is being considered for application in areas such as security monitoring and medical image analysis to improve the accuracy of lesion detection and public safety. In precision agriculture, the model can be used to monitor crop health, detect pests and diseases, and automate harvesting. In environmental monitoring, the model can be used to detect and classify different types of pollution or signs before natural disasters.

## Supporting information

**S1 File.**
(DOC)

## Author Contributions

**Investigation:** Malathy Batumalay.

**Methodology:** Xiaoli Wang.

**Project administration:** Xiaoli Wang.

**Resources:** Malathy Batumalay.

**Supervision:** Siti Sarah Maidin.

**Validation:** Siti Sarah Maidin.

**Writing – original draft:** Siti Sarah Maidin.

**Writing – review & editing:** Xiaoli Wang, Malathy Batumalay.

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
