## [Decision Letter · Decision Letter 0]

8 Oct 2024

PONE-D-24-40368PCB Defect Detection Based on Pseudo-inverse Transformation and YOLOv5PLOS ONE

Dear Dr. Wang,

Thank you for submitting your manuscript to PLOS ONE. After careful consideration, we feel that it has merit but does not fully meet PLOS ONE’s publication criteria as it currently stands. Therefore, we invite you to submit a revised version of the manuscript that addresses the points raised during the review process.

 Please revise your manuscript on the basis of the reviewers' comments. Moreover, it is suggested to emphasis more on the novelty of your work and make a comparison of your work with other previously reported ones.

We look forward to receiving your revised manuscript.

Kind regards,

Azim Uddin, Ph.D.

Academic Editor

PLOS ONE

Journal Requirements: When submitting your revision, we need you to address these additional requirements. 1. Please ensure that your manuscript meets PLOS ONE's style requirements, including those for file naming. The PLOS ONE style templates can be found at https://journals.plos.org/plosone/s/file?id=wjVg/PLOSOne_formatting_sample_main_body.pdf and https://journals.plos.org/plosone/s/file?id=ba62/PLOSOne_formatting_sample_title_authors_affiliations.pdf 2. Please note that PLOS ONE has spec6ific guidelines on code sharing for submissions in which author-generated code underpins the findings in the manuscript. In these cases, all author-generated code must be made available without restrictions upon publication of the work. Please review our guidelines at https://journals.plos.org/plosone/s/materials-and-software-sharing#loc-sharing-code and ensure that your code is shared in a way that follows best practice and facilitates reproducibility and reuse. 3. We note that Figure 6 in your submission contain copyrighted images. All PLOS content is published under the Creative Commons Attribution License (CC BY 4.0), which means that the manuscript, images, and Supporting Information files will be freely available online, and any third party is permitted to access, download, copy, distribute, and use these materials in any way, even commercially, with proper attribution. For more information, see our copyright guidelines: http://journals.plos.org/plosone/s/licenses-and-copyright. We require you to either (1) present written permission from the copyright holder to publish these figures specifically under the CC BY 4.0 license, or (2) remove the figures from your submission: A. You may seek permission from the original copyright holder of Figure 6 to publish the content specifically under the CC BY 4.0 license.  We recommend that you contact the original copyright holder with the Content Permission Form (http://journals.plos.org/plosone/s/file?id=7c09/content-permission-form.pdf) and the following text:“I request permission for the open-access journal PLOS ONE to publish XXX under the Creative Commons Attribution License (CCAL) CC BY 4.0 (http://creativecommons.org/licenses/by/4.0/). Please be aware that this license allows unrestricted use and distribution, even commercially, by third parties. Please reply and provide explicit written permission to publish XXX under a CC BY license and complete the attached form.” Please upload the completed Content Permission Form or other proof of granted permissions as an ""Other"" file with your submission.  In the figure caption of the copyrighted figure, please include the following text: “Reprinted from [ref] under a CC BY license, with permission from [name of publisher], original copyright [original copyright year].” B. If you are unable to obtain permission from the original copyright holder to publish these figures under the CC BY 4.0 license or if the copyright holder’s requirements are incompatible with the CC BY 4.0 license, please either i) remove the figure or ii) supply a replacement figure that complies with the CC BY 4.0 license. Please check copyright information on all replacement figures and update the figure caption with source information. If applicable, please specify in the figure caption text when a figure is similar but not identical to the original image and is therefore for illustrative purposes only.

Reviewers' comments:

Reviewer's Responses to Questions

**Comments to the Author**

1. Is the manuscript technically sound, and do the data support the conclusions?

Reviewer #1: Partly

Reviewer #2: Partly

2. Has the statistical analysis been performed appropriately and rigorously? 

Reviewer #1: No

Reviewer #2: Yes

3. Have the authors made all data underlying the findings in their manuscript fully available?

Reviewer #1: No

Reviewer #2: Yes

4. Is the manuscript presented in an intelligible fashion and written in standard English?

Reviewer #1: Yes

Reviewer #2: No

5. Review Comments to the Author

Reviewer #1: Comments on manuscript ID PONE-D-24-40368:

Title: “PCB Defect Detection Based on Pseudo-inverse Transformation and YOLOv5”

Some key suggestions are listed below:

1. There are certain grammatical and misspelling issues noted throughout the text, like “TOLOv5 is further improved using Transformer”. Please check and make sure to address them.

2. The abstract is not well written. The abstract could benefit from a more concise presentation of key points. Some sentences feel lengthy and complex, which can reduce readability.

3. The key contributions of your proposed method are missing in introduction section.

4. The methodology should explain how the PCB-DD dataset was constructed, including details about the number of samples, types of defects included, resolution of images, and data augmentation techniques. This information is critical for replicability and for understanding the dataset’s role in enhancing model performance. It would be useful to compare the proposed PCB-DD dataset against existing publicly available datasets (if any exist) to demonstrate the model's applicability in broader scenarios.

5. The conclusion can benefit from a broader reflection on the potential impact of the proposed model beyond the experiment. How might this technology affect industries that rely on PCB production? What future applications, besides just PCB-DD, could benefit from the advancements in the model?

Reviewer #2: Title: “PCB Defect Detection Based on Pseudo-inverse Transformation and YOLOv5”

Some key suggestions are:

1. Some misspellings and grammatical issues noted throughout the text. Please check and make sure to address them all.

2. The term "pseudo-inverse transformation" is used without prior context. Since it is a key component of the method, a brief explanation or rationale for choosing this approach would enhance understanding for readers unfamiliar with this technique.

3. The mention of "Transformer is introduced to improve YOLOv5" is quite brief. It would be helpful to provide a bit more insight into how the Transformer architecture is incorporated and why it enhances the performance of YOLOv5 in this specific context.

4. The abstract can be reorganized slightly to follow a more typical structure: problem statement, method, results, and significance.

5. The methodology does not explain how PIT is applied or why it was chosen over other traditional methods like Fourier transforms or wavelet-based methods. Including a brief explanation of why PIT is particularly well-suited for PCB image restoration would strengthen the methodology.

6. Dataset details are insufficient

7. The proposed methodology experimental results are not compared with any state of the art methods.

6. PLOS authors have the option to publish the peer review history of their article (what does this mean?). If published, this will include your full peer review and any attached files.

Reviewer #1: No

Reviewer #2: No

---

## [Author Response · Author response to Decision Letter 0]

21 Nov 2024

The manuscript has finished the revise.

I can confirm: "All relevant data are within the paper and its Supporting Information files"

Figure 7 has been modified and provided specific link to these images on Wikipedia

---

## [Decision Letter · Decision Letter 1]

26 Nov 2024

PCB Defect Detection Based on Pseudo-inverse Transformation and YOLOv5

PONE-D-24-40368R1

Dear Dr. Wang,

We’re pleased to inform you that your manuscript has been judged scientifically suitable for publication and will be formally accepted for publication once it meets all outstanding technical requirements.

Kind regards,

Azim Uddin, Ph.D.

Academic Editor

PLOS ONE

Additional Editor Comments (optional):

Reviewers' comments:

Reviewer's Responses to Questions

**Comments to the Author**

1. If the authors have adequately addressed your comments raised in a previous round of review and you feel that this manuscript is now acceptable for publication, you may indicate that here to bypass the “Comments to the Author” section, enter your conflict of interest statement in the “Confidential to Editor” section, and submit your "Accept" recommendation.

Reviewer #1: All comments have been addressed

Reviewer #2: All comments have been addressed

2. Is the manuscript technically sound, and do the data support the conclusions?

Reviewer #1: Yes

Reviewer #2: Yes

3. Has the statistical analysis been performed appropriately and rigorously? 

Reviewer #1: Yes

Reviewer #2: Yes

4. Have the authors made all data underlying the findings in their manuscript fully available?

Reviewer #1: Yes

Reviewer #2: (No Response)

5. Is the manuscript presented in an intelligible fashion and written in standard English?

Reviewer #1: Yes

Reviewer #2: (No Response)

6. Review Comments to the Author

Reviewer #1: All the required changes are addressed properly in the revised version. I am satisfied now with the revised version now and agree to accept.

Reviewer #2: All suggested comments have been carefully addressed, and detailed responses are provided in the revised manuscript.

7. PLOS authors have the option to publish the peer review history of their article (what does this mean?). If published, this will include your full peer review and any attached files.

Reviewer #1: No

Reviewer #2: No

---

## [Editor Report · Acceptance letter]

2 Dec 2024

PONE-D-24-40368R1 

PLOS ONE

Dear Dr. Wang, 

I'm pleased to inform you that your manuscript has been deemed suitable for publication in PLOS ONE. Congratulations! Your manuscript is now being handed over to our production team.

Kind regards, 

on behalf of

Dr. Azim Uddin 

Academic Editor

PLOS ONE